# The German validation of the Body Perception Questionnaire-Short Form (BPQ-SF) and its relation to current self-report measures of interoception

**Sebastian Brand**[1], **Markus Roman Tünte**[2,3], **Michael Witthöft**[4], **Stefanie Hoehl**[2], **Mathias Weymar**[5,6], **Carlos Ventura-Bort**[5]*

1 Clinical Psychology and Psychotherapy of Childhood and Adolescence, Johannes Gutenberg University, Mainz, Germany, 2 Department of Developmental and Educational Psychology, Faculty of Psychology, University of Vienna, Wien, Austria, 3 Vienna Doctoral School Cognition, Behavior and Neuroscience, University of Vienna, Wien, Austria, 4 Clinical Psychology, Psychotherapy and Experimental Psychopathology, Johannes Gutenberg University, Mainz, Germany, 5 Department of Biological Psychology and Affective Science, Faculty of Human Sciences, University of Potsdam, Potsdam, Germany, 6 Faculty of Health Sciences Brandenburg, University of Potsdam, Potsdam, Germany

☯ These authors contributed equally to this work.
* ventura@uni-potsdam.de

**Data Availability Statement:** Data is available via OSF server (https://osf.io/6s2qf/).

## Abstract

The Body Perception Questionnaire-Short Form (BPQ-SF) is one of the most used questionnaires to assess interoception. Although the BPQ-SF has been translated into different languages, there is no validated German questionnaire adaptation so far. Furthermore, empirical evidence outlining how the BPQ-SF relates to novel theories of interoception that distinguish between different facets, such as attention and accuracy, is still missing. The current study therefore aims at (1) validating the German version of BPQ-SF (with data from three different sites, $N_{total} = 1292$) and (2) relating it to the constructs of interoceptive accuracy and attention. In line with the original validation of the BPQ-SF, in the German version, an adequate three-factor structure and good internal consistency were found. However, reliability indexes varied between good, for the supra-and subdiaphragmatic scales and poor, for the body awareness scale. The three scales of the BPQ-SF showed significant associations with interoceptive measures and, more importantly, were related to both interoceptive accuracy and attention, suggesting that they mirror a mixture of both constructs. Finally, in relation to measures of psychopathology, the supra-and subdiaphragmatic (but not the body awareness) scales were positively related to alexythimic, anxious, and depressive tendencies. These findings provide evidence for the usability of the German version of the BPQ-SF and further shed light on the heterogeneity of the existing self-report measures of interoception.

**Funding:** M.R.T. and S.H. were funded by the Austrian Science Fund (FWF) [P33486]. The funders had no role in study design, data collection and analysis, decision to publish, or preparation of the manuscript.

**Competing interests:** The authors have declared that no competing interests exist.

## Introduction

The ability to perceive and process internal bodily signals, such as heart rate or gastrointestinal changes, is defined as interoception [1]. Interoceptive abilities have been related to various psychological processes and clinical tendencies (e.g., [1–12]). They are typically classified based on the nature of the interoceptive measure: behavioral, self-report, or their correspondence [4]. One of the most used self-report measures to evaluate interoception is the Body Perception Questionnaire-Short Form (BPQ-SF; [13, 14]). Since its introduction, the BPQ-SF has been widely used and has contributed to extensive literature on how interoception influences human behavior and experience [4, 15]. New approaches to conceptualizing and defining interoception highlight the need to distinguish between different facets of interoceptive processing within its domain (e.g., behavioral, self-report) [12, 16]. However, empirical evidence outlining how the BPQ-SF relates to these novel theories of interoceptive processing is still missing. Further, although the English version was introduced in 1993 [13], the psychometric validation of the German version of the BPQ-SF has not yet been carried out yet. Tackling these gaps in the literature, the current study aims to validate the German BPQ-SF and situate it within a recent theoretical framework of interoceptive processing [12].

In light of the novel theoretical perspectives that call for a nuanced taxonomy of interoception beyond the nature of the measurement [1, 12], recent research has aimed at shedding light on the relationship between different measures of interoception in the behavioral [17] and self-report domain [15]. In a recent study, Desmedt and colleagues [15] first systematically reviewed the existing literature to identify the five most used self-report measurements of interoception which were the BPQ-SF, the Multidimensional Assessment of Interoceptive Awareness (MAIA; [18, 19]), the Body Awareness Questionnaire (BAQ; [20]), the Private subscale of the Body Consciousness Questionnaire (PBCS; [21]), and the Self-Awareness Questionnaire (SAQ; [22]). After that, these five questionnaires' overall factorial and network structure was examined in a large sample. Results showed low convergence between questionnaires, indicating that these self-report measures tap into different constructs of interoception.

These results follow recent proposals emphasizing the distinction between different interoceptive constructs within the self-report (and behavioral) domain [12]. One such proposal is the 2 x 2 factorial model of interoception, which distinguishes between interoceptive accuracy, understood as correctly perceiving the actual state of one's body, and interoceptive attention, defined as the degree to which a person attends or focuses on bodily changes. The distinction between self-reported interoceptive accuracy (assessed with the Interoceptive Accuracy Scale, IAS; [23]) and attention (measured with the Interoceptive Attention Scale, IATS; [24]) has been confirmed in recent studies showing that these two constructs are differentially associated with clinical outcomes, including anxiety, depression, alexithymia or somatoform symptomatology [25, 26]. Interestingly, the most used questionnaires of interoception, as identified by Desmedt et al. [15], were created before the new proposals of interoception emerged.

Considering the importance of providing validated measures and the novel theoretical proposals of interoception, the current study aimed to validate the German version of the BPQ-SF and assess its fit within the 2 x 2 factorial model [12]. Furthermore, the relationship between the BPQ-SF subscales and psychopathological tendencies such as anxiety, depression and alexithymia was examined. The BPQ-SF comprises three subscales conceptualized based on the organization of the autonomic neural pathways according to the polyvagal theory [27–29]. The first scale, the Body Awareness scale, describes the sensibility to bodily signals arising from the afferent pathways. The other two, the Supradiaphragmatic and Subdiaphragmatic scales, assess the perceived reactivity from the ventral and dorsal vagal complex, which regulate organs above (e.g., face, head and visceral organs such as armpits) and below (e.g., visceral organs

related to the urinary or digestive systems) the diaphragm. In the original English and subsequent Spanish versions, the BPQ-SF scales showed good internal consistency, and each scale showed an adequate model fit for a one-factor solution. Since its development, the BPQ-SF has successfully been cross-culturally validated [30–32], but only recently the relationship between the 2x2 factorial model of interoception and the BPQ-SF subscales have been examined, mostly focusing on the body-awareness scale [e.g., 24]. These studies have generally found a positive association between the body awareness scale scores and the IATS scores, but the association with IAS scores has varied across studies [24, 30].

Following previous studies, we expected to find three different factors according to each subscale with at least an acceptable model fit, good internal consistency, and reliability indexes. In line with prior studies [23, 24, 30], we further expected that the BPQ-SF scales would show convergent validity with self-report measures of interoception. Furthermore, to explore the extent to which the subscales of the BPQ-SF are related to the constructs of self-reported interoceptive attention and accuracy, canonical correlational analysis (CCA) was performed. CCA allows for testing the association between sets of variables rather than looking at single correlations. Although single correlations inform about the relationship between variables, separately, they may provide an oversimplified understanding of the complex relation between the existing constructs [33]. Avoiding such simplification is particularly relevant in interoception research as it can contribute to a more comprehensive and global interpretation of the relationship between different interoception measures.

## Method

### Transparency and openness

This project emerged as a part of a broader multi-site collaboration [25, 26]. Data collection from most sites (https://aspredicted.org/K2L_8TW) and data analysis for the current project were preregistered (https://aspredicted.org/HC1_D39). Data is available via OSF server (https://osf.io/6s2qf/). Data for the current project was used in other publications [25, 26, 34].

### Participants

**Site 1.** 400 German-speaking individuals recruited via the online platform prolific (https://www.prolific.co/) took part in the study (data collected between 1st July, 2021 and 1st July, 2022). Participants were monetarily compensated for participation (approximately 3.50 €). If participants did not report a high proficiency in German, were below 18 or above 70 years old, or responded too fast or slow (SoSciSurvey variables DEG_TIME >100 or TIME_-RSI >2; [35]), they were excluded from the analysis. The final sample comprised 388 participants (216 women, 166 men, 6 non-binary).

**Site 2.** The sample consisted of 523 students who underwent a battery of questionnaires administered via Sona Systems (https://www.sona-systems.com/; data collected between November, 15th, 2020 and February 6th, 2022). Participants who did not report a high proficiency in German, were below 18 or above 70 years old or completed the survey double as fast as the sample mean time of completion ($N = 43$) were excluded from the analysis [35], resulting in a final sample of 480 participants (341 women, 78 men, 1 non-binary).

**Site 3.** The initial sample consisted of 506 individuals recruited via social media from the German general population (data collected from March 21st, 2020 to June 21st, 2020). Participants who did not report a high proficiency in German or were below 18 or above 70 years of age were excluded, resulting in a total sample of $N = 484$ (349 women, 130 men, 5 non-binary).

Slightly different batteries of questionnaires were administered to each site (see *N*s for each questionnaire in the Questionnaires subsection). The local ethics committees granted each site's Data collection approval.

### Ethic statement

Data collection was approved by the Ethics Committee of each psychological institute and/or university. Participants signed and informed consent prior to the participation of the study.

### Questionnaires

**Body Perception Questionnaire-Short Form (BPQ-SF) and Very Short Form (BPQ-VSF).** The BPQ-SF is composed of 46 items scored on a 5-point Likert scale, ranging from (1) never to (5) always, grouped in three subscales, targeting two main domains: Body Awareness and autonomous nervous system reactivity (ANSR; [13, 14]). The Body Awareness subscale (26 items) assesses the proportion of time a person reports being aware of bodily sensations. The ANSR domain is measured with the supra- (15 items) and subdiaphragmatic (6 items) reactivity subscales. The supradiaphragmatic subscale quantifies the body reactions above the diaphragm, and the subdiaphragmatic scale measures the gastrointestinal functions below the diaphragm. One item ('*I feel like vomiting*') is shared among both ANSR subscales.

As an alternative, a shorter measure of the body awareness domain was tested by Cabrera and colleagues [14]: the Body Perception Questionnaire-Very Short Form (BPQ-VSF). The BPQ-VSF comprises 12 items from the body awareness subscale of the BPQ-SF. The authors observed that the BPQ-VSF was a reliable measure of body awareness (see also [32]). In the current study, the BPQ-SF was administered to Site 1 and Site 2 samples, whereas Site 3 sample used only the BPQ-VSF. Therefore, Site 1 and Site 2 were used as validation samples for the BPQ-SF and BPQ-VSF, whereas Site 3 was used only as a replication sample for the BPQ-VSF. Of note, some of the items of the BPQ-VSF administered to the Site 3 sample had a slightly alternative linguistic construction (but identical meaning) to those used in the BPQ-SF (e.g., *Völlegefühl oder Aufgeblähtsein* was formulated as *voller oder geblähter Magen*). In the Tables A and B in S2 Text the German version of the BPQ-SF is provided.

**Interoceptive Accuracy Scale (IAS).** The IAS was created to measure interoceptive accuracy within the 2 x 2 factorial model of interoception [23, 25]. The questionnaire consists of 21 items rated on a 5-point Likert scale. In the current study, the IAS showed good internal consistency (validation sample: $\omega$ = .88; replication sample, $\omega$ = .85. All participants completed the IAS, *n* = 1292.

**Interoceptive Attention Scale (IATS).** Within the 2 x 2 factorial model of interoception [12], the IATS was designed to evaluate self-reported attention to interoceptive perceptions such as hunger or breathing [24]. The questionnaire compromises 21 items on a 5-point Likert scale. In the current study, the IATS showed good internal consistency, $\omega$ = .90. A total of *n* = 642 participants completed the IATS.

**Multidimensional Assessment of Interoceptive Awareness Version-2 (MAIA-2).** The MAIA-2 [19, 36] focuses on the evaluation of multiple dimensions of interoception throughout its 37 items divided into 8 scales. The 8 subscales are Noticing (4 items), Not-Distracting (6 items), Not-Worrying (5 items), Attention Regulation (7 items), Emotional Awareness (5 items), Self-Regulation (4 items), Body Listening (3 items), and Trust (3 items). Each item is rated on a 6-point Likert scale. Overall, in the current study, the subscales of the MAIA-2 showed acceptable to good internal consistency: (validation/replication): $\omega_{Noticing}$ = .72/.62, $\omega_{NonDistracting}$ = .85/.82, $\omega_{NotWorrying}$ = .80/.78, $\omega_{AttentionRegulation}$ = .83/.82, $\omega_{EmotionalAwareness}$ = .82/.80, $\omega_{SelfRegulation}$ = .84/.84, $\omega_{BodyListening}$ = .84/.76, and $\omega_{Trust}$ = .86/.79. ldiation sample:

$\omega_{Noticing}$ = .70, $\omega_{NonDistracting}$ = .83, $\omega_{NotWorrying}$ = .78, $\omega_{AttentionRegulation}$ = .82, $\omega_{EmotionalAwareness}$ = .82, $\omega_{SelfRegulation}$ = .84, $\omega_{BodyListening}$ = .82, and $\omega_{Trust}$ = .83. The MAIA-2 was completed *by n* = 1098 participants.

## Interoceptive Confusion Questionnaire (ICQ)

The ICQ [37] consists of 20 items, evaluating the difficulties interpreting one's non-affective physiological states, such as hunger, muscle pain, or arousal (e.g., '*I often find that I'm suddenly very thirsty*'). The ICQ is scored on a 5-point Likert scale. In our samples, the internal consistency of the ICQ was acceptable, $\omega$ = .66. A total of 226 participants completed the ICQ.

**Toronto Alexithymia Scale (TAS-20).** Alexithymia was assessed with the German version [38] of the TAS-20 [39], which consists of 20 items rated on a 5-point Likert scale grouped into three subscales: Difficulty Identifying Feelings (7 items), Difficulties Describing Feelings (5 items) and Externally Oriented Thinking (8 items). In the current study, the TAS-20, completed by 614 participants, showed good internal consistency, $\omega$ = .85.

**Patient Health Questionnaire 15-item version (PHQ-15) and the Patient Health Questionnaire 9-item version (PHQ-9).** Somatic symptoms were measured using the PHQ-15 [40], and depressive symptoms were assessed using the PHQ-9 [41]. Both questionnaires are part of a German screening procedure for assessing psychological complaints in individuals (PHQ-D; [42]). The PHQ-15 consists of 15 items assessing the degree of individual somatic symptoms (e.g., abdominal pain) on a 3-point scale. The PHQ-9 consists of 9 items measuring the degree of individual distress caused by depressive symptoms. The respondents indicate to which extent they are burdened by symptoms such as sadness or hopelessness on a 4-point scale. The PHQ-15 and PHQ-9 showed good internal consistency, PHQ-15: $\omega$ = .78, PHQ-9: $\omega$ = .88. The PHQ-15 and PHQ-9 were completed by $n$ = 484 participants.

**The Beck Depression Inventory (BDI-II).** The BDI-II [43, 44] measures the severity of depressive symptoms. It consists of 21 statements assessing the presence of psychological (e.g., feelings of guilt) and physiological (e.g., loss of energy) symptoms of major depression. Statements are assigned point values reflecting the severity of depressive symptoms. The BDI-II showed good internal consistency in the current sample, $\omega$ = .91. Overall, $n$ = 226 participants completed the BDI-II.

**Anxiety Sensitivity Inventory 3 (ASI-3).** The Anxiety Sensitivity Inventory-3 (ASI-3; [45, 46]) measures anxiety sensitivity, a construct referring to a person's fear of their physiological anxiety-related arousal response. The ASI-3 consists of 18 items rated on a 5-point Likert scale. In the current sample, the ASI-3 also showed good internal consistency, $\omega$ = .88. Overall, $n$ = 226 participants completed the ASI-3.

**State-Trait-Anxiety Inventory (STAI).** The STAI [47] intends to measure both state (S) and trait anxiety (T). In the current study, only the trait subscale was used. This subscale consists of 20 items rated on a 4-point Likert scale. The STAI-T showed good internal consistency in the present samples, $\omega$ = .93. Overall, $n$ = 226 participants completed the STAI-T.

## Statistical analysis

The software IBM SPSS Statistics [48], Mplus [49], and R version 4.0.5 [50] were used to perform the statistical analyses. Within R, we used the packages *tydiverse*, *psych*, *lavaan*, *lme4*, *cca*, and *cocor* [51–56].

Preliminary analyses were conducted to test for differences between Site 1 and Site 2 samples in BPQ-SF subscale and BPQ-VSF total scores. No differences between samples were observed in any of the subscales of the BPQ-SF (Bonferroni-corrected $p$-value = .017, all $p$'s > .03, Cohen's $d$ < .14). Although, for the BPQ-VSF, differences between samples arose (Site 1 vs

Site 2, $t_{776.68}$ = 2.67, $p$ = .007, $d$ = 0.1), these were rather small ($M_{site1}$ = 35.9 and $M_{site2}$ = 37.7), indicating that both samples had comparable BPQ-related scores and could be collapsed. Therefore, the demographic characteristics, convergent validity, and internal consistency were tested separately for the validation (Site 1 and Site 2 samples collapsed) and replication samples. Similarly, the factor structure of the BPQ-SF subscales (only validation sample) and the BPQ-SVF (validation and replication samples) were performed for each sample, separately.

**Confirmatory analysis of the structure of the Body Perception Questionnaire—(Very) Short Form (BPQ[V]-SF).** Following the validation methodology outlined by Cabrera and colleagues [14], the scores of items in the BPQ-(V)SF were dichotomized, with responses indicating no assignment receiving a score of zero and all other responses receiving a score of one. A predetermined three-factor solution based on established knowledge [14] was applied to the BPQ-SF. This solution comprised factors related to *body awareness* (items 1–26), *supradiaphragmatic reactivity* (items 27–41), and *subdiaphragmatic reactivity* (items 41 to 46). Similarly, for the BPQ-VSF, a one-factor solution (utilizing 12 items from the BPQ-SF) was fitted independently for both the validation and replication samples. Confirmatory factor analysis was executed employing the robust mean and variance-adjusted weighted least squares (WLSMV) procedure, which is recommended for models dealing with discrete data [57]. $\chi^2$-tests were performed to evaluate the model fit for the confirmatory factor solution. Recognizing the $\chi^2$-value's sensitivity to influence from large sample sizes, supplementary metrics, including the Root Mean Square Error of Approximation (RMSEA), the Comparative Fit Index (CFI), and the Tucker-Lewis Index (TLI) were also considered following previous research [58, 59].

**Descriptive characteristics.** Descriptive characteristics of the BPQ-SF, including mean, standard deviation, skewness, and kurtosis, are reported in the Table A in S1 Text. Correlational analysis was performed to investigate the association between age and the BPQ-SF subscales and the BPQ-VSF scores. Gender differences in BPQ-SF subscales and BPQ-VSF scores were evaluated using unpaired t-tests.

**Internal consistency and test-retest reliability.** Internal consistency was computed through the application of McDonald's omega. To assess test-retest reliability, a subset of participants from the Potsdam sample ($n$ = 57) underwent retesting. Participants were allowed to freely sign up for a retest after completing the online session without an initial time limit between the initial and retest sessions. However, only participants who completed the retests within 200 days after the initial session were considered for analysis.

Test-retest reliability was calculated using Pearson's and Spearman's correlation coefficients and the Intraclass Correlation Coefficient (ICC). Given the considerable variability in the number of days between the test and retest sessions (up to 200 days), an examination was conducted to determine whether the number of days between these sessions moderated the test-retest relationship. This analysis involved multiple regressions, with the scores at time 1 (i.e., test) as the criterion, and the scores at time 2 (i.e., retest), the time elapsed between time 1 and time 2 (in days), and their interaction as predictors.

**Convergent validity with interoceptive-related and other questionnaires.** Convergent validity was examined by testing the relationship between the BPQ-SF subscales and BPQ-VSF with other interoception scales (MAIA-2 subscales, IAS, IATS, ICQ) as well as with psychopathology-related scales (TAS-20, PHQ9, PHQ15, STAI-T, ASI-3, and BDI-II). Convergent validity was assessed using Pearson's correlations.

**Canonical correlation analysis.** To investigate how the BPQ-SF scales relate to different facets of interoception proposed in the 2x2 factorial model of interoception, we focused on the distinction between interoceptive accuracy and attention [12]. We conducted a canonical correlation analysis in R using the CCA package [56], with one factor comprising the IAS and

IATS, and the other was composed of the BPQ-SF subscales using the collapsed Site 1 and Site 2 samples.

## Results

### Demographic data

Age was not related to the Body Awareness subscale ($r = -.01$, $p = .660$), but a significant, negative correlation was found with the Supradiaphragmatic subscale ($r = -.11$, $p = .002$). The subdiaphragmatic scale was not significantly related to age ($r = .01$, $p = .84$), whereas the BPQ-VSF showed a positive relationship with age in the replication but not in the validation sample, validation sample: $r = -.01$, $p = .803$, replication sample: $r = .16$, $p < .001$.

To test for gender differences in each subscale, participants who self-identified as nonbinary were excluded from the analysis due to the small group size ($n = 12$). Although gender differences did not reach significance (Bonferroni-corrected $p$-value = .016) for the Body Awareness subscale, $t_{451.53} = 2.02$, $p = .044$, $d = 0.15$ ($M_{women} = 82.7$, $M_{men} = 79.8$), significant differences emerged in the supradiaphragmatic, $t_{468.24} = 3.11$, $p = .002$, $d = 0.24$ ($M_{women} = 23.6$, $M_{men} = 22.0$), and subdiaphragmatic subscales, $t_{551.53} = 5.51$, $p < .001$, $d = 0.40$ ($M_{women} = 11.2$, $M_{men} = 9.7$), indicating higher scores for women than men. For the BPQ-VSF, gender differences were found in the validation sample, $t_{452.7} = 2.04$, $p = .041$, $d = 0.16$ ($M_{women} = 37.3$, $M_{men} = 35.8$) but not in the replication sample, $t_{222.66} = 0.01$, $p = .99$, $d = 0.01$ ($M_{women} = 39.7$, $M_{men} = 39.7$).

### Confirmatory factor analysis

In line with established knowledge regarding potential factor solutions, we employed confirmatory factor analysis to apply a three-factor model to the Body Perception Questionnaire-Short Form (BPQ-SF) and a one-factor model to the Body Perception Questionnaire-Very Short Form (BPQ-VSF) for validation and replication samples separately. The corresponding fit indices for each analysis are presented in Table 1. According to the criteria outlined by Schermelleh-Engel and colleagues [59], all solutions demonstrated a good fit, as evidenced by the RMSEA. The CFI and TLI were noted to approach the acceptable threshold of .95 in the three-factor solution for the BPQ-SF. The BPQ-VSF one-factorial solutions signified an acceptable fit ($> .95$). Furthermore, in the short versions, we observed robust modelling of the factor structure in the replication sample, as evidenced by a nonsignificant $\chi^2$-test. The factor loadings for each version of the questionnaire and sample are detailed in Table 2.

### Internal consistency and test-retest reliability

The Body Awareness subscale showed excellent internal consistency, $\omega = .93$, and the Supra-, $\omega = .87$, and Subdiaphragmatic subscales showed good internal consistency, $\omega = .82$. Similar

**Table 1. Model fit for the BPQ-SF and BPQ-VSF.**

| Model/Index | $\chi^2$ | df | p | RMSEA | CFI | TLI |
|---|---|---|---|---|---|---|
| 3-Factor BPQ-SF | 1578.54 | 985 | < .001 | .027 (90% CI [.025, .030] | .946 | .943 |
| 1-Factor BPQ-VSF (validation) | 134.41 | 54 | < .001 | .043 (90% CI [.034, .052] | .969 | .963 |
| 1-Factor BPQ-VSF (replication) | 68.85 | 54 | .084 | .024 (90% CI [.001, .039] | .962 | .954 |

RMSEA = Root Mean Square Error of Approximation, CFI = Comparative Fit Index, TLI = Tucker Lewis Index.

**Table 2. Standardised factor loadings from confirmatory factor analysis for the BPQ-SF and BPQ-VSF.**

| Factor Loadings | | BPQ-SF | BPQ-VSF (validation) | BPQ-VSF (replication) |
|---|---|---|---|---|
| Body Awareness | Item 1 | .470 | - | - |
| | Item 2 | .584 | - | - |
| | Item 3 | .612 | .552 | .639 |
| | Item 4 | .757 | .713 | .781 |
| | Item 5 | .734 | - | - |
| | Item 6 | .629 | - | - |
| | Item 7 | .702 | .747 | .741 |
| | Item 8 | .845 | - | - |
| | Item 9 | .803 | .792 | .642 |
| | Item 10 | .673 | .717 | .450 |
| | Item 11 | .504 | - | - |
| | Item 12 | .770 | .811 | .734 |
| | Item 13 | .883 | .842 | .749 |
| | Item 14 | .854 | .866 | .400 |
| | Item 15 | .632 | - | - |
| | Item 16 | .729 | - | - |
| | Item 17 | .817 | .804 | .706 |
| | Item 18 | .689 | - | - |
| | Item 19 | .636 | - | - |
| | Item 20 | .551 | - | - |
| | Item 21 | .592 | - | - |
| | Item 22 | .693 | .733 | .638 |
| | Item 23 | .789 | - | - |
| | Item 24 | .656 | .638 | .762 |
| | Item 25 | .716 | .671 | .660 |
| | Item 26 | .779 | - | - |
| Supdiaphragmatic | Item 27 | .763 | - | - |
| | Item 28 | .558 | - | - |
| | Item 29 | .557 | - | - |
| | Item 30 | .634 | - | - |
| | Item 31 | .728 | - | - |
| | Item 32 | .689 | - | - |
| | Item 33 | .853 | - | - |
| | Item 34 | .639 | - | - |
| | Item 35 | .630 | - | - |
| | Item 36 | .594 | - | - |
| | Item 37 | .734 | - | - |
| | Item 38 | .670 | - | - |
| | Item 39 | .669 | - | - |
| | Item 40 | .615 | - | - |
| | Item 41 (Sup) | .330 | - | - |
| Subdia-phragmatic | Item 41 (Sub) | .403 | - | - |
| | Item 42 | .676 | - | - |
| | Item 43 | .781 | - | - |
| | Item 44 | .810 | - | - |
| | Item 45 | .853 | - | - |
| | Item 46 | .730 | - | - |

*Note.* According to suggested thresholds [60], factor loadings larger than .45 are marked in *italics* (indicating fair loadings), indexes above .6 are marked **bold** (indicating good loadings), and indexes below .45 are left unformatted (indicating poor loadings).

internal consistency was observed for the BPQ-VSF (validation sample: ω = .87, replication sample: ω = .87).

A total of $n$ = 57 participants completed the BPQ-SF a second time to test for test-retest reliability. The Body Awareness scale of the BPQ-SF showed poor test-retest reliability ($r$ = .47, $r_s$ = .45, ICC = .46). Whereas the supradiaphragmatic and subdiaphragmatic subscales showed good test-retest reliability (Supradiaphragmatic: $r$ = .79, $r_s$ = .73, ICC = .79; Subdiaphragmatic: $r$ = .77, $r_s$ = .71, ICC = .77). Like the Body Awareness scale, the BPQ-VSF showed poor test-retest reliability ($r$ = .53, $r_s$ = .52, ICC = .52).

Multiple regressions were performed to test the effects of time passed on the relation between test and retest scores. For the Body Awareness subscale of the BPQ-SF, the scores at time 2 significantly predicted the scores at time 1, β = 0.69, $t$ = 3.72, $p$ < .001, 95% $CI$ [0.32, 1.07], but no evidence for an effect of days passed, β = 0.32, $t$ = 1.93, $p$ = .058 95% $CI$ [−0.01, 0.66], or interaction was found, β = −0.01, $t$ = -1.77, $p$ = .080, 95% $CI$ [−0.01, 0.01]. Similar results were observed for the Supradiaphragmatic subscale, as indicated by a significant effect of scores at time 2, β = 0.78, $t$ = 7.53, $p$ < .001, 95% $CI$ [0.57, 0.99]. Still, no evidence for the effects of days passed, β = 0.04, $t$ = 0.34, $p$ = .734, 95% $CI$ [−0.07, 0.10], or the interaction, β = -0.01, $t$ = -0.82, $p$ = .420, 95% $CI$ [−0.01, 0.01]. However, for the Subdiaphragmatic subscale, along with a significant effect of scores at time 2, β = 0.97, $t$ = 7.61, $p$ < .001, 95% $CI$ [0.71, 1.22], significant effects of days passed, β = 0.05, $t$ = 2.4, $p$ = .020, 95% $CI$ [0.01, 0.12], as well as interaction effects were found, β = -0.01, $t$ = -2.04 $p$ = .047, 95% $CI$ [-0.01,- 0.01].

Finally, for the BPQ-VSF, a significant effect of scores at time 2 was observed, β = 0.92, $t$ = 4.65, $p$ < .001, 95% $CI$ [0.52, 1.32], as well as significant effects of days passed, β = 0.19, $t$ = 2.19, $p$ = .030, 95% $CI$ [0.02, 0.35], and days passed x score at time 2 interaction effects, β = -0.01, $t$ = -2.14 $p$ = .038, 95% $CI$ [-0.01,- 0.01].

## Convergent validity with interoception-related questionnaires

The complete list of correlations between BPQ-SF subscales, BPQ-VSF, and other measurements is depicted in Table 3. To test the significance of the relationship, a Bonferroni correction was applied ($p$ = .05/13 questionnaires = .003).

**Associations with interoceptive accuracy (IAS).** Body Awareness scores were positively related to IAS scores $r$ = .308, $p$ < .001, 95% $CI$ [.255, .361]. A similar relationship was observed with the BPQ-VSF validation sample, $r$ = .306, $p$ < .001, 95% $CI$ [.253, .358], and replication sample, $r$ = .439, $p$ < .001, 95% $CI$ [.376, .497]. In contrast, Supradiaphragmatic scores were negatively related to self-reported interoceptive accuracy, $r$ = -.210, $p$ < .001, 95% $CI$ [-.260, -.150].

**Associations with interoceptive attention (IATS).** Subjective interoceptive attention scores were positively related to all subscales of the BPQ and the BPQ-VSF. Body Awareness: $r$ = .198, $p$ < .001, 95% $CI$ [.135, .26], supradiaphragmatic: $r$ = .320, $p$ < .001, 95% $CI$ [.265, .382], subdiaphragmatic: $r$ = .330, $p$ < .001, 95% $CI$ [.275, .391], BPQ-VSF, $r$ = .195, $p$ < .001, 95% $CI$ [.132, .257].

**Associations with the Multidimensional Assessment of Interoceptive Awareness Version-2 (MAIA-2) subscales.** The Body Awareness subscale was positively associated with the Attention Regulation, $r$ = .160, $p$ < .001, 95% $CI$ [.093, .223], Emotional Awareness, $r$ = .260, $p$ < .001, 95% $CI$ [.200, .325], and Self-Regulation scale, $r$ = .170, $p$ < .001, 95% $CI$ [.110, .24,], as well as Body Listening scale, $r$ = .140, $p$ < .001, 95% $CI$ [.070, .200].

A similar pattern was observed for the BPQ-VSF. In the validation sample, positive relationships with the Attention Regulation, $r$ = .185, $p$ < .001, 95% $CI$ [.119, .248], Emotional Awareness, $r$ = .230, $p$ < .001, 95% $CI$ [.230, .353], and Self-Regulation scale, $r$ = .190, $p$ < .001, 95%

**Table 3. Correlational analysis between BPQ-SF subscales, BPQ-VSF, and other measurements.**

| Pearson's r | 1 | 2 | 3 | 4 |
|---|---|---|---|---|
| 1. BPQ (N = 808) | — | | | |
| 2. BPQ SUPRA (N = 808) | .171*** | — | | |
| 3. BPQ SUB (N = 808) | .216*** | .544*** | — | |
| 4. BPQ-VSF (N = 808/484) | .965*** | .168*** | .212*** | — |
| 5. IAS (N = 808) | .312*** | -.202*** | -.101** | .309*** / .439*** |
| 6. IATS (N = 642) | .207*** | .336*** | .337*** | .202*** |
| 7. MAIA-2 NOTICING (N = 614) | .275*** | -.017 | .067 | .290*** / .432*** |
| 8. MAIA-2 NONDIST (N = 614) | .059 | -.003 | -.049 | .061 / .039 |
| 9. MAIA-2 NOTWO (N = 614) | -.049 | -.220*** | -.158*** | -.033 / -.035 |
| 10. MAIA-2 ATT REG (N = 614) | .157*** | -.209*** | -.137*** | .182*** / .282*** |
| 11. MAIA-2 EMO AWA (N = 614) | .268*** | .040 | .094 | .298*** / .341*** |
| 12. MAIA-2 SELFREG (N = 614) | .176*** | -.119** | -.047 | .194*** / .180*** |
| 13. MAIA-2 BODY LIS (N = 614) | .139*** | -.017 | .039 | .151*** / .253*** |
| 14. MAIA-2 TRUST (N = 614) | .079 | -.263*** | -.211*** | .096* / .101* |
| 15. ICQ (N = 226) | -.188** | .330*** | .166* | -.205** |
| 16. TAS-20 DESC (N = 614) | -.038 | .230*** | .111** | -.065 |
| 17. TAS-20 IDENT (N = 614) | -.005 | .395*** | .265*** | -.029 |
| 18. TAS-20 EXT (N = 614) | -.150*** | .071 | -.014 | -.156*** |
| 19. TAS-20 (N = 614) | -.077 | .309*** | .166*** | -.102* |
| 20. PHQ9 (N = 484) | — | — | — | .041 |
| 21. PHQ15 (N = 484) | — | — | — | .083 |
| 22. STAI-T (N = 226) | -.069 | .363*** | .245*** | -.075 |
| 23. ASI-3 (N = 226) | .056 | .383*** | .278*** | .021 |

(*Continued*)

**Table 3.** (Continued)

| Pearson's *r* | 1 | 2 | 3 | 4 |
|---|---|---|---|---|
| 24. BDI-II<br>(*N* = 226) | -.058 | .378*** | .274*** | -.051 |

Note

* p < .05

** p < .01

*** p < .001

Supra = Supradiaphragmatic, Sub = Subdiaphragmatic, IAS = Interoceptive Accuracy Scale, IATS = Interoceptive Attention Scale, MAIA2 = Multidimensional Assessment of Interoceptive Awareness Version 2 (Subscales in order Noticing, Non-Distracting, Not-Worrying, Attention Regulation, Emotional Awareness, Self-Regulation, Body-Listening, Trust), ICQ = Interoceptive Confusion Questionnaire, TAS-20 = Toronto Alexithymia Scale (Subscales in order Describing Feelings, Identifying Feelings, Externally Oriented Thinking), BDI-II = Beck Depression Inventory II, STAI-T = Trait Anxiety Inventory, ASI-3 = Anxiety Sensitivity Inventory. Values of both the validation and the replication sample were provided (validation/replication) when possible.

CI [.128, .256], as well as Body Listening scale, $r$ = .149, $p$ < .001, 95% CI [.083, .214], were observed. This pattern was also found in the replication sample, for Attention Regulation, r = .280, p < .001, 95% CI [.213, .352], Emotional Awareness, $r$ = .350, p < .001, 95% CI [.280, .412], and Self-Regulation scales, $r$ = .181, $p$ < .001, 95% *CI* [.108, .253], as well as Body Listening scales, $r$ = .256, $p$ < .001, 95% CI [.185, .325].

On the other hand, the Supra- and Subdiaphragmatic scales showed negative associations with the Not Worrying (supradiaphragmatic: $r$ = -.210, $p$ < .001, 95% CI [-.277, -.149], subdiaphragmatic: $r$ = -.154, $p$ < .001 (95% CI [-.220, -.089]), Attention Regulation (supradiaphragmatic: $r$ = -.210, $p$ < .001, 95% CI [-.268, -.140], subdiaphragmatic: $r$ = -.130, $p$ < .001, 95% CI [-.199, -.067]), and Trust scales (supradiaphragmatic: $r$ = -.250, $p$ < .001, 95% CI [-.310, -.190], subdiaphragmatic: $r$ = -.210, $p$ < .001, 95% CI [-.268, -.141]).

**Interoceptive Confusion Questionnaire (ICQ).** The ICQ showed negative relationships with Body Awareness, $r$ = -.180, $p$ < .001, 95% CI [-.291, -.078], and BPQ-VSF, $r$ = -.210, $p$ < .001, 95% CI [-.309, -.098], but was positively correlated to the Supradiaphragmatic scale, $r$ = .320, $p$ < .001, 95% CI [.218, .416].

**Toronto Alexithymia Scale-20 (TAS-20).** The overall scores of the TAS-20 were positively related to supra-, $r$ = .300, $p$ < .001, 95% CI [.233 –.356], and subdiaphragmatic scale scores, $r$ = .160, $p$ < .001, 95% CI [.091, .220].

**Psychopathological symptoms.** Scores in the BDI-II were positively related to supra-, $r$ = .360, $p$ < .001, 95% CI [.260, .452], and subdiaphragmatic scale scores, $r$ = .250, $p$ < .001, 95% CI [.139, .35]. Similarly, both scales were positively related to the STAI-T, supradiaphragmatic, $r$ = .350, $p$ < .001, 95% CI [.249, .443], subdiaphragmatic, $r$ = .230, $p$ < .001, 95% CI [.118, .327], and ASI-3 scores, supradiaphragmatic, $r$ = .380, $p$ < .001, 95% CI [.277, .466], subdiaphragmatic, $r$ = .270, $p$ < .001, 95% CI [.164, .368].

**Canonical correlation analysis.** Canonical Correlation Analysis revealed significant relationships between both factors. Two canonical variates were extracted; a canonical variate is a composite variable created from a weighted combination of the original variables within each set, such that the correlation between these composites from each set is maximized. We found that both canonical variates showed a significant canonical correlation: first canonical correlation: 0.407 (Wilks' λ = 0.703, Rao's $F$(6, 1274) = 40.98, p < .001), second canonical correlation: 0.397 (Wilks' λ = 0.842, Rao's $F$(2, 638) = 59.70, $p$ < .001).

Next, we investigated the canonical loadings. Table 4 displays the canonical loadings for each variable within the canonical variates. One canonical variate loaded strongly on the IAS,

**Table 4. Canonical loadings.**

| Variable | Canonical Variate 1 | Canonical Variate 2 | Group |
|---|---|---|---|
| IAS | -1.000 | -0.021 | Factor 1 |
| IATS | -0.008 | -1.000 | Factor 1 |
| BPQ Awareness | -0.956 | -0.316 | Factor 2 |
| BPQ Supra | 0.531 | -0.495 | Factor 2 |
| BPQ Sub | 0.082 | -0.492 | Factor 2 |

*Note*: IAS = Interoceptive Accuracy Scale; IATS = Interoceptive Attention Scale; BPQ = Body Perception Questionnaire; Canonical loadings represent the correlation between the variables and their respective canonical variate. Loadings closer to -1 or 1 indicate a stronger relationship.

indicating it captures variance related to an interoceptive accuracy dimension, while the second canonical variate loaded strongly on the IATS, suggesting that it captures variance associated with an interoceptive attention dimension. The BPQ-SF subscales had significant loadings across both dimensions, indicating that they tap into both constructs captured by the canonical variates (see also Fig 1).

## Discussion

The current study aimed to validate the German version of the BPQ-(V)SF, one of the most used interoception questionnaires [15] and examine its fit within new models of interception. In line with the initial [14] and subsequent adaptation to other languages [30–32], in the German version of the BPQ-SF, the three-factor solution showed a good model fit. The three subscales further revealed good to excellent internal consistency. As predicted, subscales of the BPQ-SF were significantly associated with other measures of interoception, but such relationships varied across scales. Furthermore, CCA indicated that the subscales of the BPQ-SF do not specifically relate to interoceptive accuracy or attention but load on both facets.

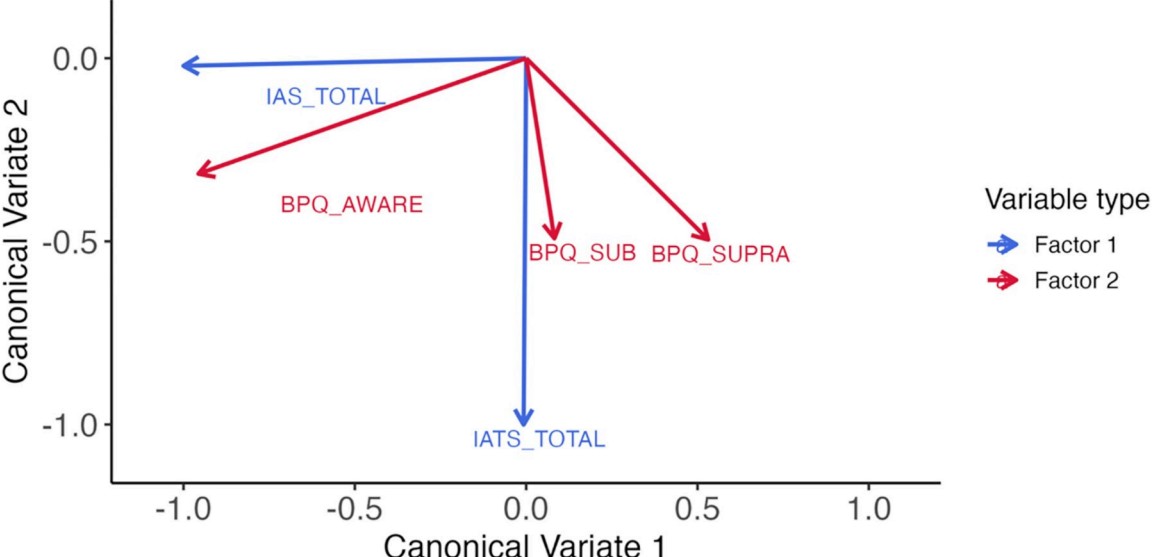

**Fig 1. Canonical variates.** *Note*: IAS = Interoceptive Accuracy Scale; IATS = Interoceptive Attention Scale; BPQ_AWARE = Body Perception Questionnaire Awareness Subscale; BPQ_SUB = Subdiaphragmatic Scale; BPQ_SUPRA = Supdiaphragmatic Subscale.

Confirmatory factor analysis revealed the three-factor solution to be a good fit for the German version of the BPQ-SF. These results align with previous validations of the BPQ-SF [30–32] and further support the theoretical conceptualization of the questionnaire. In addition, the BPQ-VSF also demonstrated an excellent one-factor solution, replicating previous findings [14, 31]. It should be noted that, as for other versions of the BPQ-SF, in the German version not all items loaded optimally to the three factors according to current guidelines [60], suggesting that the questionnaire may benefit from further refinement. Although all scales showed good to excellent internal consistency, the good test-retest reliability observed for the Supra- and Subdiaphragmatic scales contrasted with the poor reliability scores observed for the Body Awareness scale and the BPQ-VSF. These findings suggest that self-reported autonomic reactivity might reflect more stable constructs than self-reported awareness of bodily changes, which may be more influenced by short-term, fluctuating factors [26].

In relation to demographic characteristics, a significant negative relationship between age and the Supra- and Subdiaphragmatic scales was observed. These results align with previous validations of the BPQ-SF [14, 31] and concur with findings suggesting decreased sensibility of autonomic reactivity with age [61]. We also observed lower autonomic reactivity scores in men than women, partly replicating previous findings [31]. Although no specific hypotheses were derived in this regard, gender differences in performance tasks [62, 63] and self-reported measures of interoception have often [26, 31] but not always [25] been found. However, these findings have pointed to both increased [26, 62] and decreased interoceptive processing for women compared to men [63]. Our results thus add to the mixed results on the relation between gender and interoception.

As expected, scales of the BPQ-(V)SF showed significant associations with other self-report measures of interoception. However, the direction and significance of these associations varied across scales. Whereas the Body Awareness subscale and the BPQ-SF showed positive, albeit small, associations with most of the interoceptive scales, the Supra- and Subdiaphragmatic subscales showed nonsignificant or negative associations with interoceptive questionnaire scales (except for a positive association with the IATS). To shed more light on how the BPQ-SF subscales fit the recently proposed 2 x 2 factorial model of interoception [12], CCA was performed, using self-reported interoceptive accuracy and attention as canonical variates. Results from the CCA indicate that the BPQ-SF subscales are multidimensional, loading on interoceptive accuracy and attention factors. These results contradict prior studies that assumed that the Body Awareness subscale reflects self-reported interoceptive attention (e.g., [30]). Instead, the Body Awareness scales may mirror a mixture of both constructs. Our results are thus in line with previous findings [15], highlighting the lack of consistency between the existing self-report measures of interoception and ratifying the need for more comprehensive frameworks that encompass the different interoceptive constructs measured with the current self-report measures [1, 15, 16].

In relation to questionnaires of psychopathological symptomatology, the BPQ-VSF and Body Awareness scale showed nonsignificant or negative associations. In contrast, the Supra and Subdiaphragmatic scales were positively related to alexithymic, anxious, and depressive tendencies. These findings are partly in line with previous results showing a positive association between autonomic reactivity and depression scores [32]. In a similar vein, Kolacz and colleagues [64] have recently observed a positive association between the autonomic reactivity scales of the BPQ-SF and disrupted physiological reactivity, including reduced heart rate variability, a psychophysiological readout typically associated with psychopathological vulnerability [65, 66]. Recent proposals have emphasized the role of disrupted autonomic functioning in clinical disorders (e.g., [67]). The current findings may thus provide indirect evidence for

these proposals, demonstrating the negative role of higher autonomic reactivity on psycho-pathological tendencies.

Two main implications can be extracted from our results. First, improving our understanding of the ways in which existing self-report measures of interoception are related can help to clarify the taxonomy of interoception in a more comprehensive and inclusive manner. Second, highlighting the associations between different facets of interoception (e.g., autonomic reactivity) and psychopathological tendencies, may promote the development of intervention programs [68] that tackle specific interoceptive abilities to improve mental health.

## Conclusion

In summary, the current findings provide evidence for the usability of the German version of the BPQ-(V)SF, extending previous questionnaire versions. By examining the relationship with other interoceptive and psychopathology-related questionnaires, the current study further shed light on the heterogeneity of the existing self-report measures of interoception and their differential relationship to psychopathology tendencies [26, 69], supporting the need for more comprehensive interoceptive frameworks [16].

## Supporting information

**S1 Text.**
(DOCX)

**S2 Text.**
(DOCX)

## Acknowledgments

We thank P.B. for her valuable contribution to the Site 3 sample data collection as part of her master thesis. We thank J. P. W. for his valuable contribution in the German translation of the BPQ-SF.

## Author Contributions

**Conceptualization:** Sebastian Brand, Markus Roman Tünte, Carlos Ventura-Bort.

**Data curation:** Sebastian Brand, Markus Roman Tünte, Carlos Ventura-Bort.

**Formal analysis:** Sebastian Brand, Markus Roman Tünte, Carlos Ventura-Bort.

**Investigation:** Sebastian Brand, Markus Roman Tünte, Carlos Ventura-Bort.

**Methodology:** Markus Roman Tünte, Carlos Ventura-Bort.

**Project administration:** Carlos Ventura-Bort.

**Supervision:** Michael Witthöft, Stefanie Hoehl, Mathias Weymar.

**Writing – original draft:** Sebastian Brand, Markus Roman Tünte, Carlos Ventura-Bort.

**Writing – review & editing:** Sebastian Brand, Markus Roman Tünte, Michael Witthöft, Stefanie Hoehl, Mathias Weymar, Carlos Ventura-Bort.

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
