## [Decision Letter · Decision Letter 0]

28 Feb 2024

PMEN-D-24-00030

The German validation of the Body Perception Questionnaire-Short Form (BPQ-SF) and its relation to current self-report measures of interoception

PLOS Mental Health

Dear Dr. Ventura-Bort,

Thank you for submitting your manuscript to PLOS Mental Health. After careful consideration, we feel that it has merit but does not fully meet PLOS Mental Health’s publication criteria as it currently stands. Therefore, we invite you to submit a revised version of the manuscript that addresses the points raised during the review process.

We look forward to receiving your revised manuscript.

Kind regards,

Jinjin Lu, Ph.D.

Academic Editor

PLOS Mental Health

Journal Requirements:

https://journals.plos.org/mentalhealth/s/figures 

https://journals.plos.org/mentalhealth/s/figures#loc-file-requirements 

Additional Editor Comments (if provided):

Please follow the reviewers’ comments and revise it.

Reviewers' comments:

Reviewer's Responses to Questions

**Comments to the Author**

1. Does this manuscript meet PLOS Mental Health’s publication criteria? Is the manuscript technically sound, and do the data support the conclusions? The manuscript must describe methodologically and ethically rigorous research with conclusions that are appropriately drawn based on the data presented.

Reviewer #1: Yes

Reviewer #2: Yes

2. Has the statistical analysis been performed appropriately and rigorously?

Reviewer #1: Yes

Reviewer #2: Yes

3. Have the authors made all data underlying the findings in their manuscript fully available (please refer to the Data Availability Statement at the start of the manuscript PDF file)?

Reviewer #1: Yes

Reviewer #2: Yes

4. Is the manuscript presented in an intelligible fashion and written in standard English?

Reviewer #1: Yes

Reviewer #2: Yes

5. Review Comments to the Author

Reviewer #1: This is an interesting and easy to read study. The authors have collected a unique dataset using a sound methodology. The paper is generally well written and structured.

However,

- I feel this unique dataset has not been utilised or discussed to its full extent.

- The role of co-authors may be included.

Reviewer #2: The paper has been well-written.

The manuscript lacks key words.

All indices must have a specified threshold, especially Table 2 requires captions.

Selecting a loading option below is not appropriate.What is 60? The goal of this work is to validate a questionnaire in the German language. However, the loadings are inadequate for validating the questionnaire.

Authors should aim to clarify the cross loadings, Principal Component Analysis (PCA), and Exploratory Factor Analysis (EFA) for questionnaire validation.

The study's implications should also be provided.

6. PLOS authors have the option to publish the peer review history of their article (what does this mean?). If published, this will include your full peer review and any attached files.

**Do you want your identity to be public for this peer review?** For information about this choice, including consent withdrawal, please see our Privacy Policy.

Reviewer #1: No

Reviewer #2: No

---

## [Editor Report · Decision Letter 1]

11 Apr 2024

The German validation of the Body Perception Questionnaire-Short Form (BPQ-SF) and its relation to current self-report measures of interoception

PMEN-D-24-00030R1

Dear Dr. Ventura-Bort,

We are pleased to inform you that your manuscript 'The German validation of the Body Perception Questionnaire-Short Form (BPQ-SF) and its relation to current self-report measures of interoception' has been provisionally accepted for publication in PLOS Mental Health.

Best regards,

Jinjin Lu, Ph.D.

Academic Editor

PLOS Mental Health